# Immunomodulatory Effect of Isocaloric Diets with Different Protein Contents on Young Adult Sprague Dawley Rats

**DOI:** 10.3390/foods12081597

**Published:** 2023-04-10

**Authors:** Liuting Wu, Rui An, Yuyue Xi, Zhiru Tang, Tiejun Li, Yetong Xu, Jiaman Pang, Xie Peng, Weizhong Sun, Zhihong Sun

**Affiliations:** 1Laboratory for Bio-Feed and Molecular Nutrition, College of Animal Science and Technology, Southwest University, Chongqing 400715, China; 17752785935@163.com (L.W.); xiyuyue0202@163.com (Y.X.); tangzr@swu.edu.cn (Z.T.); xyt8501@163.com (Y.X.); pangim@swu.edu.cn (J.P.); pengxie2022@swu.edu.cn (X.P.); swz2012@swu.edu.cn (W.S.); 2Institute of Animal Husbandry Science, Sichuan Academy of Agricultural Sciences, Chengdu 610066, China; 13271925710@163.com; 3Institute of Subtropical Agriculture, The Academy of Chinese Natural Sciences, Changsha 410125, China; tjli@isa.ac.cn

**Keywords:** SD rats, dietary protein content, immune cells, TLR4/NF-κB, immunological status

## Abstract

To understand the potential mechanisms of dietary protein on intestinal and host health, we studied the immunomodulatory effects of isocaloric diets with high or low crude protein (CP) contents on young adult Sprague Dawley (SD) rats. A total of 180 healthy male rats were randomly assigned to six groups (six replicate pens per treatment with five rats per pen) and fed diets with 10% CP, 14% CP, 20% CP (control), 28% CP, 38% CP, and 50% CP. Compared with the control diet, the rats fed the 14% CP diet significantly elevated lymphocyte cell counts in the peripheral blood and ileum, whereas the 38% CP diet significantly activated the expression of the TLR4/NF-κB signaling pathway in the colonic mucosa (*p* < 0.05). Moreover, the 50% CP diet reduced growth performance and fat deposition and increased the percentages of CD4^+^ T, B, and NK cells in the peripheral blood and the colonic mucosal expression of IL-8, TNF-α, and TGF-β. Overall, rats fed the 14% CP diet enhanced host immunity by increasing the numbers of immune cells, and the immunological state and growth of SD rats were negatively impacted by the diet containing 50% CP.

## 1. Introduction

The immune system is made up of immune cells, immune organs, and inflammatory cytokines, with the intestine serving as an important immune organ in animals [1]. Several immunological processes occur in the mucosa of the intestine, and the intestinal mucosal barrier serves as an initial barrier against the external environment [2]. Therefore, maintaining host intestinal health is essential for nutrient digestion and absorption and the prevention of intestine-related disorders. The intestine contains the largest number of immune cells, including lymphocytes and monocytes, among the organs in the body, and is constantly exposed to several antigens and possible immunological triggers [3]. Lymphocytes are well-acknowledged as key immune system components, acting as peripheral sentinels capable of rapidly mobilizing protective tissue immunity upon pathogen recognition [4]. Abnormalities in the interactions between lymphocytes and their subsets can alter the initiation of inflammatory responses, thereby exacerbating disease conditions [5].

Previous studies have shown a close link between dietary sugar and lipid intake and immunological state and illness in animals [6]. Moreover, changes in protein intake have been reported to alter the levels of these metabolites in the intestine [7]. These changes can disrupt the tightly regulated intestinal microenvironment, increasing the host’s susceptibility to intestinal inflammatory diseases [8]. It has been found that both the high-protein diet (HPD) [9] and the low-protein diet (LPD) [10] stop fat from building up and improve the nutritional status in obese patients. Although a LPD may reduce inflammation in people with chronic kidney disease [11], it appears that a very LPD worsens inflammation and malnutrition in rats with uremic renal disease [12]. Additionally, in diabetic patients, HPD lowers insulin resistance [13], whereas feeding HPD to healthy rats can damage the structural integrity of the intestinal mucosal layer of the intestinal epithelium, promoting inflammatory response in the host [14]. Therefore, it is crucial to describe the effects of HPDs and LPDs in healthy animals and their potential disease-related mechanisms. However, appropriate dietary protein concentration could improve the immunological status of animals suffering from inflammatory diseases (review, [11]).

Although the effects of dietary protein contents on gut and host health have been examined previously, most studies focused on diets with a single high or low protein content. Therefore, the study mainly investigated the effects of three HPD and two LPD with varying levels of casein on the growth performance, intestinal mucosa barrier, and immunological status of SD rats, to elucidate the potential mechanisms of dietary protein on intestinal and host health.

## 2. Materials and Methods

### 2.1. Animal Use and Care

The studies carried out in this work were conducted in strict accordance with Chinese guidelines for animal welfare. All animal treatments were performed in compliance with the Guidelines for Care and Use of Laboratory Animals of Southwest University and were approved by the Animal Ethics Committee of Southwest University (Chongqing, China). A total of 180 young adult Sprague Dawley (SD) male rats (six weeks old) were purchased from Chongqing Medical University and housed under the following conditions for 6 weeks: temperature, 20–22 °C; humidity, 60 ± 5%. The animals had unrestricted access to drinking water.

### 2.2. Experimental Diets and Design

After 7 days of adaption, 180 young adult rats were randomly assigned to six dietaries regimens, with six replicates (five rats per replicate) per treatment group. The crude protein (CP) contents of the six isocaloric diets were 10%, 14%, 20% (control), 28%, 38%, and 50%. The experimental period lasted 42 d. The control diet was designed in accordance with the instructions from the Standardization Administration of China (SAC). Table 1 and Appendix A show the ingredients and chemical composition of the six experimental diets.

### 2.3. Recording and Sample Collection

The initial and final weights of the rats were measured, and daily feed intake was recorded throughout the experimental period. Apart from the body weight gain and feed-to-gain ratio, which were displayed as a graph in Figure 1 in the study, the other data about growth performances are reported in the Appendix A Appendix A. At the conclusion of the 6-week trial, one rat was randomly selected from each replicate, starved for 12 h, and then sedated with 10% chloral hydrate (3 mL/kg). After administering anesthesia, 2 mL of whole blood was collected from the abdominal vein and sent to the laboratory for routine blood test. Additionally, samples of 5 mL of blood were obtained from the abdominal vein into tubes without anticoagulants, held at room temperature (about 25 °C) for 2 h, centrifuged at 3000 rpm for 15 min, and the supernatant was collected in 1.5 mL sterile centrifuge tubes and stored at −80 °C for physiological and biochemical analyses. The carcasses of the executed rats were placed on a crushed ice bed for tissue collection. The rats were dissected and the morphology of the main organs of the rat was observed. The weights of the liver, spleen, kidney, thymus, epididymis fat, and bilateral perirenal fat were measured, and the corresponding organ indices were calculated. The middle part of the ileum and colon (3 cm each) were fixed for morphological assessment. The middle segment of the ileum and colon (10 cm each) were rinsed with normal saline, and mucosa were scraped with sterile glass slides. For further analysis, the ileum and colon mucosa were immediately frozen in liquid nitrogen and preserved in a freezer at −80 °C.

### 2.4. Chemical Analysis

Dietary chemical compositions were assessed using the methods of Association of Official Analytical Chemists. 2005 [12]. Using DAO and D-Lac enzyme-linked immunosorbent (ELISA) assay kits (CV < 15%; Enzymatic Biotechnology Co., Ltd., Shanghai, China), the DAO and D-Lac activities in serum were determined, according to the manufacturer’s protocol.

Fresh colonic mucosa from the rats were homogenized in a 1:10 (*m*/*v*) dilution of radioimmunoprecipitation assay (RIPA) lysis buffer (Beyotime, Nanjing, China). The concentrations of inflammatory cytokines, including (TNF)-α, (TGF)-β, immunoglobulin (Ig)G, IgM, sIgA, interleukin (IL)-6, IL-8, and IL-10, were determined in the homogenates using enzyme-linked immunosorbent assay (ELISA) kits (inter-assay CV < 15%, Enzymatic Biotechnology Co., Ltd., Shanghai, China).

Ileum and colon samples were fixed in 4% formalin. After rinsing with water, the samples were dehydrated using graded concentrations of alcohol (50%, 70%, 80%, 90%, and 100%). Multiple sections (4 µm) were deparaffinized using xylene and stained with hematoxylin and eosin (H&E) stain for general histological examination. Hematoxylin and eosin stain images for general histological analysis are presented in Appendix A.

### 2.5. Western Blotting

TLR4/NF-κB proteins expression were evaluated using western blot analysis. Briefly, to collect the supernatants, colonic mucosa samples from the rats were lysed in RIPA buffer with protease inhibitors (Beyotime, Shanghai, China) and centrifuged (Thermo Fisher Scientific, Waltham, MA, USA) at 4 °C for 15 min at 5000 rpm. Subsequently, protein in the supernatants were separated using SDS-PAGE on 10% acrylamide gels under denaturing conditions, and transferred onto 0.45 μm PVDF membranes, following the manufacturer’s instructions. The membrane was blocked for 1 h at room temperature (about 25 °C) with 5% (wt/vol) skim milk in Tris-buffered saline (TBS)/Tween 20 before being incubated overnight at 4 °C with primary antibodies, including β-actin (China Peptides, Shanghai, China), Myd88 (Millipore, Temecula, CA, USA), TLR4 (Sigma, St Louis, MO, USA), NF-κB (Invitrogen, Carlsbad, CA, USA), NOD2 (HuaAn Biotechnology, Hangzhou, Zhejiang Province, China) antibodies (The dilution employed for β-actin was 1:5000, while that for TLR4, NOD2, MYD88, and NF-KB was 1:1000). Following that, the samples were incubated in a blocking solution with peroxidase-conjugated anti-rabbit IgG antibody (Beyotime, Shanghai, China) according to the manufacturer’s instructions before being visualized with a chemiluminescence reagent (Millipore Corporation, Billerica, MA, USA). The relative expression of TLR4/NF-κB proteins was quantified using ImageJ V1.8.0 software (National Institutes of Health, Bethesda, MD, USA).

### 2.6. Immunocytometric Analysis

An automatic biochemical analyzer (Sysmex XT-2000i; Tokyo, Japan) was used to determine the white blood cell, neutrophil, lymphocyte, monocyte, eosinophil, basophil counts, and the ratio of neutrophils, lymphocytes, monocytes, eosinophils, as well as basophils. Additionally, the expression of four lymphocyte subsets, including CD3^+^CD4^+^CD8^−^, CD3^+^CD4^−^CD8^+^, CD45^+^CD3^−^CD45R^+^, and CD45^+^CD3^−^CD161^hi^, were assayed. The samples were incubated with monoclonal antibodies against CD3^+^CD4^+^CD8^−^, CD3^+^CD4^−^CD8^+^, CD45^+^CD3^−^CD45R^+^, and CD45^+^CD3^−^CD161^hi^ to label helper (CD4^+^) T-cells, cytotoxic (CD8^+^) T-cells, B cells, and natural killer (NK) cells, respectively. To determine the expression of lymphocyte subsets in the peripheral blood, 2 μL of monoclonal antibody or corresponding isotype (control) was added to flow cytometric tube (Corning, Thermo Fisher Scientific, Waltham, MA, USA) containing 50 μL of blood. The sample was incubated in the dark for more than 30 min at room temperature (approximately 25 °C), followed by 10-min incubation in the dark with 1 × BD FACS lysing solution (BD Biosciences, Franklin Lake, NJ, USA). Thereafter, 2 mL of PBS was added and the solution was thoroughly mixed before centrifuging at 350 rpm for 5 min and collecting the supernatant. PBS (500 μL) was added to the supernatant and properly mixed before determining the expression levels of lymphocyte subsets using a flow cytometer (CytoFLEX, BD Biosciences, Franklin Lake, NJ, USA).

To determine the expression of lymphocyte subsets in the colon, colon samples were cut using sterile scalpel and scissors, rinsed with PBS 3–5 times, cut into small pieces of about 1 mm, and cleaned with PBS several times. Thereafter, 1 mL of collagenase (50–200 units/mL, dissolved in PBS) was added to the tissue fragment, incubated for 10 min to 1 h at 37 °C, and agitated every 5 min to completely dissociate the tissues, followed by the addition of 3% fetal bovine serum to terminate digestion. The cell suspension was filtered through sterile stainless-steel mesh or nylon mesh to separate dispersed cells, tissue fragments, and large fragments. If further depolymerization was required, fresh collagenase was added to the fragments, and the suspension was washed in PBS through centrifugation 3–5 times. Subsequently, 5 μL of monoclonal antibody or a corresponding isotype (control) was added to flow cytometric tube containing 50 μL single cell suspension and incubated in the dark at room temperature (about 25 °C) for more than 30 min, followed by further incubation in 1 × BD FACS lysing solution (BD Biosciences) in the dark for 10 min. Finally, 2 mL of PBS was added to the samples, thoroughly mixed, and centrifuged at 350 rpm for 5 min to collect the supernatant. PBS (500 μL) was added to the supernatant, thoroughly mixed, and the expression levels of lymphocyte subsets were determined using a flow cytometer (CytoFLEX, BD Biosciences, Franklin Lake, NJ, USA).

### 2.7. Statistical Analysis

The GLM program in SAS 8.1 statistical software (SAS Institute, Inc., Cary, NC, USA) was used to analyze all physiological and biochemical data through one-way analysis of variance (ANOVA), which was followed by Tukey’s test for multiple mean comparison. *p* values <0.05, <0.01, and <0.001 were determined statistically significant, moderately significant, and extremely significant, respectively.

## 3. Results

### 3.1. Growth Performance and Organ Morphology Development

Figure 1A,B shows that rats in 10% CP and 50% CP groups had lower total body weights gain (*p* < 0.05), but higher ADFI/ADG ratios (*p* < 0.05) compared with those in the control group (20% CP diet). Appendix A shows that rats in 10% CP, 28% CP, and 50% CP groups had lower body weight gain; moreover, there was a significant increase in protein intake and a significant decrease in the ratio of body weight gain to protein intake in the rats in the 28% CP, 38% CP, and 50% CP groups, and there was a significant decrease in protein intake and a significant increase in the ratio of body weight gain to protein intake in the rats in the 10% CP and 14% CP groups, compared with the control group. Additionally, there was an increase in the epididymal fat (*p* < 0.05) weight of rats in the 14% CP group compared with that of rats in the control group (Figure 1F). The kidney and spleen indices of rats fed the 28% CP diet were higher (*p* < 0.05) than those of rats in the control group (Figure 1J,M). Moreover, rats fed the 38% CP diet had higher (*p* < 0.05) thymus indices than rats in the control group (Figure 1N). However, there were no differences (*p* > 0.05) in kidney, bilateral perirenal fat, spleen, and thymus weights (Figure 1D,E,G,H) or liver, bilateral perirenal fat, and epididymal fat indices among the six groups (Figure 1I,K,L).

### 3.2. Colonic Mucosal Inflammatory Cytokines

Both rats in the LPD (10% and 14% CP diets) and HPD groups had higher (*p* < 0.05) colonic mucosal concentrations of IgG than those in the control group. Additionally, rats in the 28% CP group had higher (*p* < 0.05) colonic mucosal concentrations of IgM, sIgA, IL-6, and IL-8 compared with the control group. Moreover, rats in the 50% CP group had higher (*p* < 0.05) colonic mucosal concentrations of IL-8, TNF-α, and TGF-β than those in the control group. However, there was no significant difference (*p* > 0.05) in the colonic mucosal concentration of IL-10 among the six groups (Table 2).

### 3.3. Serum DAO and D-Lactate Concentrations and Intestinal Morphology

Rats in the HPD (28%, 38%, and 50% CP diets) groups had higher (*p* < 0.05) serum DAO concentrations than those in the control group (Table 2). Moreover, rats fed the 14% CP diet had a higher (*p* < 0.05) ileal lymphocyte population than those in the control group. Furthermore, there was a decrease (*p* < 0.05) in the crypt depth of the colon of rats in the 38% CP group compared with the control group (Table 3; Appendix A).

### 3.4. The Protein Expression of TLR4/NF-κB Signaling Pathway

There was an increase (*p* < 0.01) in TLR4 and NF-κB protein expression levels in the colonic mucosa of rats in the 38% CP group compared with the control group (Figure 2H,I). Moreover, the 28% CP diet enhanced (*p* < 0.01) NF-κB protein expression in the colonic mucosa of the rats compared with the control diet (Figure 2I).

### 3.5. Peripheral Blood Cell Counts

There was an increase (*p* < 0.05) in white blood cell and lymphocyte counts in the 14% CP group compared with the control group. Moreover, the eosinophil ratio was significantly lower (*p* < 0.05) in the 10% CP group compared with the control group. However, there were no significant differences (*p* < 0.05) in neutrophil, basophil, monocyte, and eosinophil counts and the percentages of neutrophils, basophil, lymphocyte, and monocyte among the six groups (Table 4).

### 3.6. Lymphocyte Subsets in Peripheral Blood and Colon Tissue

There was a decrease (*p* < 0.05) in the percentage of CD8^+^ T cells and an increase (*p* < 0.01) in the percentages of B and NK cells in the peripheral blood of rats in the 50% CP group compared with the control group (Figure 3E–G). Moreover, Figure 4D shows that there was an increase (*p* < 0.05) in the percentage of CD4^+^ T cells in the colon of rats in the 14% and 50% CP groups compared with the control group. However, the percentages of CD8^+^ T, B, and NK cells in the colon of the rats were not significantly affected (*p* > 0.05) by the diets (Figure 4E–G).

## 4. Discussion

In the present study, rats fed 50% CP diets for six weeks had a lower final body weight and higher ADFI/ADG than those fed the control diet, which was consistent with previous findings. For example, there was a decrease in both food intake and body weight in animals fed 30% CP diets for 2 or 4 weeks [16]. Sarr et al. (2011) found that increasing the dietary protein content contributed to a temporary reduction in the adiposity of the low-birth weight baby [17]. A high-protein diet under energy restriction maintained the muscle mass of animals and promoted adipose tissue utilization for energy production [18], resulting in a decrease in adipose tissue. Consistent with the body weight, there was a decrease in adipose tissue in rats in the 50% CP group compared with the 20% CP group, confirming that the decrease in body weight may be attributed to a decrease in fat deposition in the rats.

The immune system and pathogen defense are vital and highly energy-intensive aspects of an organism development [19]. The thymus, which produces T cells, and the spleen, which is necessary for lymphocyte recirculation and B cell maturation, are two critically vital immune organs [20]. According to our study, rats in the 38% CP group had a higher thymus index than control rats did, whereas rats in the 28% CP group had a higher spleen index. A previous study reported that atrophy of immune organs in rats fed protein-deficient diets was improved by increasing dietary protein content, indicating that immune organs are sensitive to the energy and protein content of their feed [21]. Overall, these results showed that diets containing 28% and 38% CP could enhance immune function in rats by promoting the development of immune organs.

Serum DAO and D-Lac concentrations have been linked to intestinal mucosal integrity, and a rise in these concentrations is thought to signal intestinal mucosal damage [22]. The current study showed that rats fed the 50% CP diet increased serum DAO and D-Lac concentrations. Studies have shown that feeding DAO promoters to mammals can exacerbate intestinal diseases and increase animal morbidity and mortality [23]. Moreover, D-Lac, a product of bacterial metabolism and lysis, is equally detrimental to gut health [24]. Therefore, it was suggested that feeding 50% CP diet may be detrimental to the intestinal mucosal of SD rats.

Furthermore, morphological examination of the ileum showed an increase in lymphocyte counts in the ileum of rats fed the 14% CP diet. The intestinal mucosa is composed of epithelial cells that secrete mucus and antimicrobial peptides to improve innate immunity, and a decrease in the number of epithelial cells is a potential pathogenesis of chronic mucosal inflammation [25]. Hence, this result showed that feeding the 14% CP diet to rats may improve the intestinal mucosa immunity by increasing the number of immune cells.

Our study revealed that rats in the 50% CP group showed higher colonic mucosal expression levels of IL-8, TNF-α, and TGF-β than the rats in the control group. IL-8 is important in local inflammation, and excess immune responses can increase its secretion, leading to inflammatory cell aggregation and oxidative stress [26]. TGF-β and TNF-α are essential for suppressing immune responses and are highly expressed in patients with autoimmune diseases and cancer [27], indicating that the increase in proinflammatory substances in the colonic mucosa of rats fed 50% CP diet may be linked to autoimmune disorders.

Additionally, there was an increase in TLR4 and NF-κB protein expression levels in the colonic mucosa of rats fed the 38% CP diet. In addition to serving as a barrier, intestinal epithelial cells can detect and respond to inflammatory disease through the TLR4/NF-κB signaling pathway [28]. TLR4 is believed to be part of the first immune barrier to bacterial infection in the gastrointestinal tract and can promote the release of cytokines by activating the NF-κB signaling pathway, thus activating innate immune response [29]. According to Cario (2010), upregulated NF-κB signaling pathway in the intestinal mucosa induced chronic inflammatory responses, indicating the activation of immune responses in rats fed the 38% CP diet [30].

Despite the fact that research on the effects of dietary protein content on immune cells in the body is limited, a prior study revealed that lymphocyte counts are reduced in the blood of rats with high-sugar and high-fat diet-induced obesity-associated diseases [31]. In the present study, there was an increase in lymphocyte counts in the peripheral blood of rats in the 14% CP diet group, whereas there was a decrease in eosinophil ratio in the rats in the 10% CP diet group. Eosinophils are involved in several homeostatic and inflammatory responses and are thought to have a protective effect against obesity and related diseases [32]. Van et al. (2017) observed a marked decrease in eosinophil counts in C57BL/6 mice fed a high-fat diet. According to these findings, feeding the rats the 10% CP diet might contribute to obesity and related diseases, while rats fed the 14% CP diet showed an improvement [33].

Furthermore, rats fed the 50% CP diet had a lower proportion of CD3^+^CD8^+^ T cells in the peripheral blood, and a higher proportion of CD3^+^CD4^+^ T cells in the colonic tissue compared to rats fed the control diet. An alteration in the population of lymphocyte subsets in peripheral blood can result in immune diseases. Hegazy et al. (2017) reported a decrease in the proportion of CD3^+^CD4^+^ T cells and an increase in CD3^+^CD4^+^ T cell count in the peripheral blood of patients with inflammatory bowel disease [34]. This proportional enrichment in intestinal tissues suggests that CD3^+^CD4^+^ T cells from patients with inflammatory bowel disease were recruited from the blood to the intestine to support the intestine by producing barrier-protective cytokines and providing a large pool of pathogen-specific antibodies [35,36]. In addition, patients with Crohn’s disease had a lower number of CD3^+^CD8^+^ T cells in their peripheral blood [37], which was also observed in obese patients (BMI ≥ 35) [38]. Therefore, the changes in CD8^+^ T and CD4^+^ T cell counts in the 50% CP group indicated a potential damage in the immune response in the SD rats.

Alterations in the quantity and function of B and NK cells have been associated with autoimmune diseases [39]. In the present study, the proportion of B and NK cells in the peripheral blood was higher in the 50% CP group than in the control group. For instance, adults with latent autoimmune diabetes have a higher percentage of peripheral B-lymphocyte subsets compared with healthy individuals [40]. This result indicated that the 50% CP diet inhibited the immune differentiation function of lymphocytes in rats and might result in immune dysregulation in the host. Overall, these findings provide a theoretical basis for the effects of the HPD on immune functions in rats.

## 5. Conclusions

In summary, the present study demonstrated that the rats fed the diet containing 14% CP exhibited elevated immune cell counts in their peripheral blood and intestinal tissue. Additionally, the growth of immunological organs was facilitated in rats fed diets containing 28% and 38% CP. In contrast, 10% CP and 50% CP diets reduced the growth performance, as the 50% CP diet impaired the functioning of the immune system, as evidenced by an increase in serum DAO and D-Lac concentrations; colonic mucosal expression of IL-8, TNF-α, and TGF-β; and alterations in the lymphocyte subsets in the peripheral blood (CD8+ T, B cells, and NK cells) and the colon (CD4+ T cells), which were detrimental to host health and may trigger autoimmune dysregulation.

## Figures and Tables

**Figure 1 foods-12-01597-f001:**
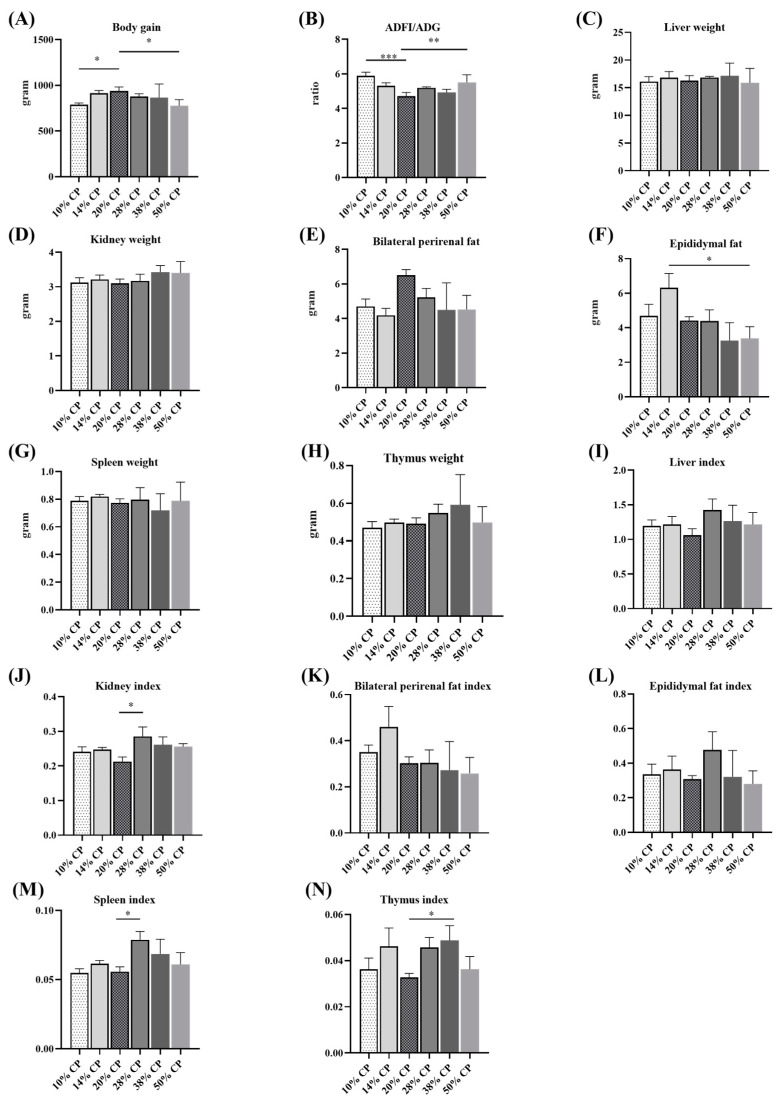
Effects of feeding diets with different protein contents for 6 weeks on body weight gain (**A**), feed-to-gain ratio (ADFI/ADG) (**B**), liver weight (**C**), kidney weight (**D**), bilateral perirenal fat weight (**E**), epididymal fat weight (**F**), thymus weight (**G**), spleen weight (**H**), and the ratios of liver (**I**), kidney (**J**), bilateral perirenal fat (**K**), epididymal fat (**L**), thymus (**M**), and spleen (**N**) relative to the body weight (*n* = 6). * *p* < 0.05, ** *p* < 0.01, and *** *p* < 0.001, compared with the 20% crude protein diet.

**Figure 2 foods-12-01597-f002:**
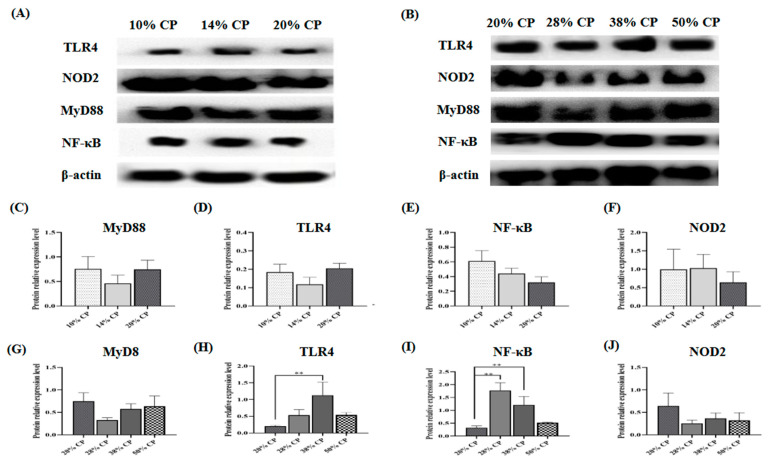
Effects of feeding diets with different protein contents for 6 weeks on SD rats on the TLR4/NF-κB signaling pathway-related protein expression. Low-protein diets (**A**), High-protein diets (**B**), Myd88 (**C**,**G**), TLR4 (**D**,**H**), NF-κB (**E**,**I**), NOD2 (**F**,**J**) in the colonic mucosal of SD rats (*n* = 4). ** *p* < 0.01, compared with the 20% crude protein diet.

**Figure 3 foods-12-01597-f003:**
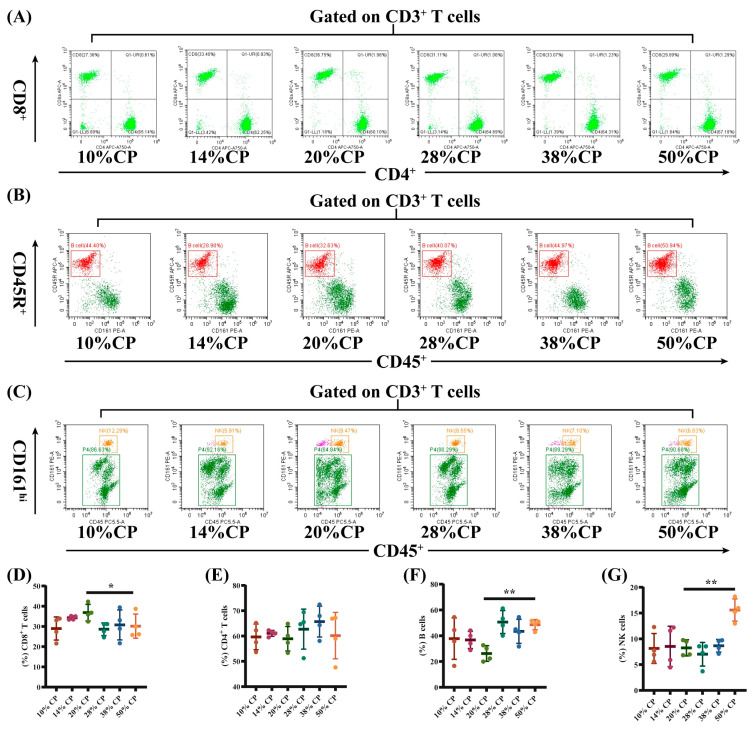
Effects of feeding diets with different protein concentrations for 6 weeks on the lymphocyte subsets in the peripheral blood of SD rats (*n* = 4). PBLS were stained with appropriate antibodies and isotype controls and initially gated on CD3^+^ T cells. Dot plots depicting the approach to gating on (**A**) CD4^+^ and CD8^+^ T cells; (**B**) B cells; and (**C**) NK cells in representative rats fed 10%, 14%, 20% (control), 28%, 38%, and 50% crude protein diets. Percentages of circulating (**D**) CD4^+^ T cells; (**E**) CD8^+^ T cells; (**F**) B cells; and (**G**) NK cells in rats fed 10%, 14%, 20% (control), 28%, 38%, and 50% crude protein diets. * *p* < 0.05 and ** *p* < 0.01, compared with the 20% crude protein diet.

**Figure 4 foods-12-01597-f004:**
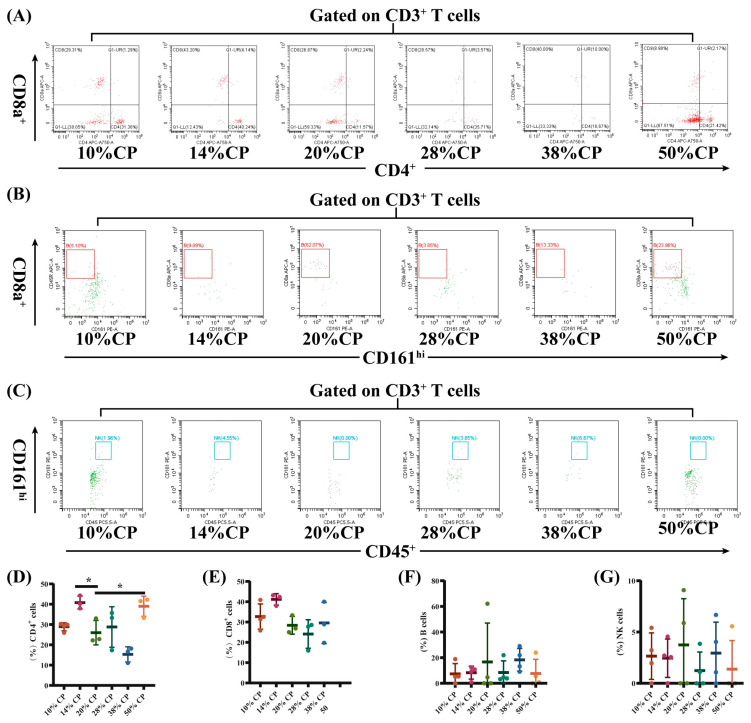
Effects of feeding diets with different protein concentrations for 6 weeks on the lymphocyte subsets in the colon tissue of SD rats (*n* = 3–4). The colon tissue lymphocyte subsets were stained with appropriate antibodies and isotype controls and initially gated on CD45^+^ and CD3^+^ T cells. Dot plots depicting the approach to gating on (**A**) CD4^+^ and CD8^+^ T cells; (**B**) B cells; and (**C**) NK cells in rats fed of 10%, 14%, 20% (control), 28%, 38%, and 50% crude protein diets. Percentages of circulating (**D**) CD4^+^ T cells; (**E**) CD8^+^ T cells; (**F**) B cells; and (**G**) NK cells in rats fed the 10%, 14%, 20% (control), 28%, 38%, and 50% CP crude protein diets. * *p* < 0.05, compared with the 20% crude protein diet.

**Table 1 foods-12-01597-t001:** Ingredients and composition of diets differing in CP contents.

Items	Treatments					
	10% CP	14% CP	20% CP	28% CP	38% CP	50% CP
Ingredients (%)						
Corn	65.35	62.64	57.36	47.80	34.15	17.81
Casein	1.90	6.90	14.28	24.31	36.94	52.13
Soybean oil	8.70	7.79	6.55	5.71	5.22	4.61
Wheat bran	9.00	8.70	9.00	9.60	10.50	11.55
Alfalfa meal	6.00	5.80	6.00	6.40	7.00	7.70
Limestone	0.70	0.68	0.61	0.53	0.43	0.29
Dicalcium phosphate	3.30	3.04	3.10	3.20	3.30	3.46
Salt	0.20	0.20	0.20	0.20	0.20	0.20
Choline chloride	0.25	0.25	0.25	0.25	0.25	0.25
Met	0.41	0.28	0.09	0	0	0
L-Lys hydrochloride	1.16	0.76	0.15	0	0	0
Thr	0.53	0.37	0.10	0	0	0
Trp	0.14	0.08	0	0	0	0
Arg	0.66	0.52	0.31	0	0	0
Premix ^1^	2.0	2.0	2.0	2.0	2.0	2.0
Total	100	100	100	100	100	100
Composition						
ME ^2^ (Kcal/kg)	3572	3572	3572	3572	3572	3572
CP ^3^	11.9	14.0	20.4	27.8	38.1	48.5
CF ^3^	3.12	3.08	3.08	3.10	3.07	3.06

^1^ Composition per kg of premix: Mg 2 g, K 5 g, Fe 10 mg, Zn 30 mg, Mn 7.5 mg, Cu 10 mg, I 0.5 mg, Se 0.15 mg, vitamin A 0.014 million IU, vitamin D3 1500 IU, vitamin E 120 IU, vitamin K 5 mg, vitamin Bl 13 mg, vitamin B2 12 mg, vitamin B6 12 mg, vitamin Bl2 0.022 mg, niacin 60 mg, pantothenic acid 24 mg, folic acid 6 mg, biotin 0.2 mg. ^2^ Calculated values according to the data of Chinese table of feed ingredients and nutritional (AOAC, 2005) [15]. Abbreviation: ME, metabolizable energy. ^3^ Determined values.

**Table 2 foods-12-01597-t002:** Effects of dietary protein contents on the intestinal damage markers and colonic cytokines of SD rats.

Items	Treatments						SEM	*p*-Value
	10% CP	14% CP	Control	28% CP	38% CP	50% CP		
**Intestinal damage markers**								
DAO (μmol/L)	1.63 ^ab^	1.57 ^ab^	1.27 ^b^	1.76 ^a^	1.68 ^a^	1.90 ^a^	1.02	0.005
D-lactate (μmol/L)	3.67 ^b^	3.72 ^b^	4.56 ^ab^	4.83 ^a^	4.38 ^ab^	4.82 ^a^	0.26	0.009
**Colonic cytokines**								
IgG (μmol/L)	2.34 ^b^	2.21 ^b^	1.88 ^c^	2.92 ^a^	2.29 ^b^	2.38 ^b^	0.11	<0.001
IgM (μmol/L)	291 ^bc^	292 ^bc^	280 ^c^	367 ^a^	308 ^ab^	338 ^ab^	13.8	<0.001
sIgA (μg/mL)	4.52 ^b^	4.18 ^bc^	3.54 ^c^	5.19 ^a^	3.79 ^bc^	4.08 ^bc^	0.23	<0.001
IL-6 (pg/mL)	33.8 ^ab^	31.8 ^b^	30.7 ^b^	38.8 ^a^	32.5 ^b^	35.0 ^ab^	1.47	0.007
IL-8 (pg/mL)	37.1 ^bc^	31.8 ^bc^	30.6 ^c^	42.3 ^a^	32.5 ^bc^	38.3 ^b^	1.44	<0.001
IL-10 (pg/mL)	11.3	12.0	11.0	12.9	11.1	11.7	0.65	0.422
TNF-α (pg/mL)	56.5 ^b^	58.3 ^b^	55.0 ^b^	62.1 ^ab^	53.0 ^b^	65.4 ^a^	2.48	0.021
TGF-β (pg/mL)	45.2 ^ab^	45.1 ^ab^	42.6 ^b^	48.6 ^ab^	40.2 ^b^	50.5 ^a^	2.09	0.010

^a,b,c^ Values within a row with different superscripts differ significantly. (*p* < 0.05). Data are presented as means. Results shown correspond to the end of the experiment (day 42). (*n* = 6). Abbreviations: CP, Crude protein; SEM, Standard error of measurement, DAO, Diamine oxidase; IgG, Immunoglobulin G; IgM, Immunoglobulin M; sIgA, Secretory immunoglobulin A; IL-6, Interleukin-6; IL-8, Interleukin-8; IL-10, Interleukin-10; TNF-α, Tumor necrosis factor-α; TGF-β, Transformation growth-β.

**Table 3 foods-12-01597-t003:** Effects of dietary protein contents on the intestinal mucosa morphology and goblet cell and lymphocyte population of SD rats.

Items	Groups						SEM	*p*-Value
	10% CP	14% CP	Control	28% CP	38% CP	50% CP		
**ileum**								
villus height (μm)	326	277	359	316	332	387	31.2	0.45
crypt depth (μm)	280	253	329	257	267	257	31.9	0.43
villus height/crypt depth	1.16	1.11	1.11	1.24	1.25	1.21	0.05	0.15
goblet cells number	36.7	20.8	21.7	23.6	27.3	30.3	4.08	0.08
lymphocytes number	57.0 ^ab^	76.2 ^a^	39.8 ^b^	45.0 ^b^	46.3 ^b^	36.3 ^b^	7.29	<0.01
**colon**								
crypt depth (μm)	109 ^a^	88.6 ^ab^	87.8 ^ab^	89.9 ^ab^	74.5 ^b^	83.5 ^ab^	6.21	0.02
goblet cells number	40.2	44.2	40.8	44.3	31.0	37.2	3.12	0.66
lymphocytes number	10.3	7.83	3.33	10.8	6.67	6.17	6.20	0.56

^a,b^ Values within a row with different superscripts differ significantly. (*p* < 0.05). Data are presented as means. Results shown correspond to the end of the experiment (day 42). (*n* = 6). Abbreviations: CP, Crude protein; SEM, Standard error of measurement.

**Table 4 foods-12-01597-t004:** Effects of dietary protein contents on the peripheral blood cell counts of SD rats.

Parameters	Treatments						SEM	*p*-Value
	10% CP	14% CP	Control	28% CP	38% CP	50% CP		
**Peripheral blood cell counts**								
WBC (×10^9^/L)	12.5 ^b^	14.4 ^a^	10.3 ^b^	12.3 ^b^	11.0 ^b^	11.6 ^b^	0.658	<0.001
NEUT (×10^9^/L)	1.91	2.36	1.49	1.92	1.59	1.69	0.247	0.226
BASO (×10^9^/L)	0.037	0.050	0.042	0.048	0.03	0.033	<0.01	0.146
LYMPH (×10^9^/L)	10.1 ^ab^	11.5 ^a^	8.38 ^b^	10.32 ^ab^	9.58 ^ab^	9.13 ^ab^	0.596	0.02
MONO (×10^9^/L)	0.40	0.43	0.42	0.48	0.43	0.32	0.076	0.905
EO (×10^9^/L)	0.028	0.045	0.048	0.05	0.048	0.048	<0.01	0.074
**Peripheral blood cell** **percentages**								
NEUT (%)	15.9	18.0	13.8	15.7	15.2	14.6	1.11	0.116
BASO (%)	0.28	0.35	0.32	0.33	0.25	0.28	0.04	0.478
LYMPH (%)	82.3	78.3	83.1	82.2	83.4	81.8	1.44	0.352
MONO (%)	3.8	3.2	4.0	3.8	4.3	0.43	0.53	0.536
EO (%)	0.22 ^b^	0.32 ^ab^	0.43 ^a^	0.42 ^a^	0.48 ^a^	0.42 ^a^	0.04	0.003

^a,b^ Values within a row with different superscripts differ significantly. (*p* < 0.05). Data are presented as means. Results shown correspond to the end of the experiment (day 42). (*n* = 6). Abbreviations: CP, Crude protein; SEM, Standard error of measurement; WBC, white blood cell; NEUT, neutrophils; BASO, basophil; LYMPH, lymphocyte; MONO, monocyte; EO, eosinophil.

## Data Availability

All related data and methods are presented in this paper. Additional inquiries should be addressed to the corresponding author.

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
