# Peer review of "Immunomodulatory Effect of Isocaloric Diets with Different Protein Contents on Young Adult Sprague Dawley Rats"

_foods, 2023, doi:10.3390/foods12081597_

Round 1

Reviewer 1 Report

Dear Editor

After carefully reviewing the manuscript: Immunomodulatory effect of isocaloric diets with different protein contents on young adult Sprague-Dawley rats ......

I want to make some comments.

Regarding the objective of the research: we mainly investigated the effects of three HPD and two low protein diets (LPD) with different levels of casein on growth performance low protein diets (LPD) with different casein levels on growth performance, intestinal mucosal barrier, and immune status of SD rats to elucidate the potential mechanisms of dietary protein on intestinal and host health.

1.- Regarding the growth results, the author does not present the results of total food intake (g), food intake (g/day), protein intake (g/day), initial body weight (g), final body weight (g), body weight gain (g)/g food, body weight gain (g)/g protein intake growth performance.

2.- About the Experimental design, the use of 180 rats; randomly allotted to six dietary treatments, with six replicates (five rats per replicate) per treatment group. The animal Ethics Committee of Southwest University ignores the principles of the 3Rs (Replacement, Reduction, and Refinement). The reduction at the minimum of the animals experimental is of essential observance for the committees. Eight hundred animals are too many. Only 36 rats in the experimental design are enough.

3.- The quality of all the graphics in all figures is so deficient. It is necessary to improve the resolution and size and limit the use of the statement: % PC

4.- The author does not report the dilution employed for each protein evaluated in Western blotting methods.

5.- The authors mentioned the sample collection and fixed staining of ileum and colon samples were fixed in 4% formalin. After rinsing with water, the samples were dehydrated using graded concentrations of alcohol (50%, 70%, 80%, 90%, 113 and 100%). Multiple sections (4 μm) were deparaffinized using xylene and stained with hematoxylin and eosin (H&E) stain for general histological examination. But not present the images of the tissues

6- Concerning Figures 2a and b, the authors present only one protein expression of the immunoblot of TLR4. NOD2, MYD88, NF-KB, and B-actin. When 36 animals were used to remove the colonic mucosal; Additionally, they were present in two different gel treatments. The correct form is to present at least three protein expressions of three animals other for each treatment in the same gel.

In addition, the authors failed to present the amino acid profile of each of the experimental diets and to evaluate and discuss the effect of excess or deficiency of each of the amino acids present in high or low-protein diets.

Reviewer 2 Report

The manuscript “Immunomodulatory effect of isocaloric diets with different  protein contents on young adult Sprague-Dawley rats” is a novel and comprehensive study.

The experimental design issufficient. Please find my minor suggestions for acceptance.

Abstract

Page 1, Lines 17-19: “Compared with the control diet, the rats fed the 14% CP diet elevated lymphocyte cell counts in the peripheral blood and ileum, whereas the 38% CP diet activated the expression of the TLR4/NF-κB signaling pathway in the colonic mucosa.” Please mention if these effects are statistically important, with p value. Here and in through the manuscript.

Introduction

I believe the introduction section is too short. Please add more details on previous studies.  

“Additionally, extremely low levels of dietary protein can exacerbate inflammation and nutritional status in rats [9], whereas feeding high protein diets (HPD) to healthy rats can damage the structural integrity of the intestinal mucosal layer of the intestinal epithelium, promoting inflammatory response in the host [10].” So how?

Page 2, Lines 70-71: How have you decided these six amounts (10%, 14%, 20% (control), 28%, 38%,70 and 50%) of  crude protein (CP) contents?

Page 11: Please rewrite the conclusion part more clearly.

Please redesign Figure 1, it is not clear and deformed.

Figure 3 and Figure 4 need to be redesigned. The graphs are not readable, deformed. Maybe you can divide into 2 different figures or give some of the data in the Supplement.

Generally, this kind of enlarging, significantly lowers the graph quality.

Reviewer 3 Report

Dear Authors, thanks for fine work, but please verify some points:

1. The section of Conclusion is missing and the authors mixed it by discussion while it should be address in separate part!  

2- It is recommended:

 In 2.4, lines 106-115, it is better to mention  inter-assay / intra-assay CV,!

[There are two types of %CVs that are used to express the precision of immunoassay results: intra-assay CV and inter-assay CV. Intra-assay CV is a measure of the variance between data points within an assay, meaning sample replicates ran within the same plate. Inter-assay CV is a measure of the variance between runs of sample replicates on different plates that can be used to assess plate-to-plate consistency. As a general guideline, to gauge the overall reliability of your immunoassay results, inter-assay %CV should be less than 15% while intra-assay %CV should be less than 10%.]

3- The quality of figures 1,[3&4 parts B and C must be more clear and legible! 

Round 2

Reviewer 1 Report

Dear Authors

Thank you for your modifications. However, the quality of the figure must improve. 

The digital photograph of the colon and ileum tissues must indicate the ocular size used.

The author has not included the gel's Western blot original. 
